# A critical role for mTORC1 in erythropoiesis and anemia

**Zachary A Knight[†‡], Sarah F Schmidt[†], Kivanc Birsoy[§¶], Keith Tan, Jeffrey M Friedman***

Laboratory of Molecular Genetics, Howard Hughes Medical Institute, The Rockefeller University, New York, United States

**\*For correspondence:** vale@ucsf.edu

[†]These authors contributed equally to this work

**Present address:** [‡]Neuroscience Graduate Program, University of California, San Francisco, San Francisco, United States; [§]Whitehead Institute for Biomedical Research, Cambridge, United States; [¶]Massachusetts Institute of Technology, Cambridge, United States

**Competing interests:** The authors declare that no competing interests exist.

**Reviewing editor**: David Ginsburg, Howard Hughes Medical Institute, University of Michigan, United States

**Abstract** Red blood cells (RBC) must coordinate their rate of growth and proliferation with the availability of nutrients, such as iron, but the signaling mechanisms that link the nutritional state to RBC growth are incompletely understood. We performed a screen for cell types that have high levels of signaling through mTORC1, a protein kinase that couples nutrient availability to cell growth. This screen revealed that reticulocytes show high levels of phosphorylated ribosomal protein S6, a downstream target of mTORC1. We found that mTORC1 activity in RBCs is regulated by dietary iron and that genetic activation or inhibition of mTORC1 results in macrocytic or microcytic anemia, respectively. Finally, ATP competitive mTOR inhibitors reduced RBC proliferation and were lethal after treatment with phenylhydrazine, an inducer of hemolysis. These results identify the mTORC1 pathway as a critical regulator of RBC growth and proliferation and establish that perturbations in this pathway result in anemia.

## Introduction

Cells must coordinate their rate of growth and proliferation with the availability of nutrients. mTOR, a serine–threonine kinase, is one the key proteins responsible for nutrient signaling in eukaryotic cells. mTOR is activated by conditions that signal energy abundance, such as the availability of amino acids, growth factors, and intracellular ATP. Activated mTOR then phosphorylates a set of downstream targets that promote anabolic processes, such as protein translation and lipid biosynthesis, while suppressing catabolic processes such as autophagy (*Zoncu et al., 2011*).

mTOR resides in two cellular complexes that have distinct functions and regulation (*Loewith et al., 2002*; *Sarbassov et al., 2004*). mTOR complex 1 (mTORC1) is sensitive to inhibition by the natural product rapamycin and contains the protein Raptor. Key targets of mTORC1 include S6 kinase (S6K), which influences cell size, and the eIF4-E binding protein (4E-BP1), which regulates cell proliferation through effects on cap-dependent translation (*Shima et al., 1998*; *Dowling et al., 2010*). mTOR complex 2 (mTORC2) is resistant to rapamycin and contains the protein Rictor. mTORC2 phosphorylates and activates several kinases in the AGC family, such as Akt and SGK, on a sequence known as the hydrophobic motif (*Sarbassov et al., 2005*). As Akt itself activates mTORC1 by phosphorylation of the Tsc1/2 complex, these two kinases reciprocally regulate each other in response to growth factor signals.

TOR was discovered in yeast, where it functions as a nutrient sensor regulating cell growth and proliferation (*Heitman et al., 1991*). This cell-autonomous function is conserved in higher organisms, and there has been rapid progress in delineating the molecular pathways by which mTOR controls basic cellular processes such as protein translation (*Zoncu et al., 2011*). Less is understood about how mTOR signaling links nutrient signals to cellular growth to regulate the physiologic state of multicellular organisms such as mammals. However, the availability of mice bearing conditional alleles of essential mTORC1 and mTORC2 components has made it possible to begin to dissect the physiologic

**eLife digest** To multiply and grow, cells need to create more of the molecules—such as proteins—that make up their structure. This only happens if the cell has a good supply of the nutrients used to build the proteins.

Red blood cells are particularly sensitive to the supply of nutrients, especially iron, which is a key component of the hemoglobin molecules that enable the cells to transport oxygen around the body. A lack of iron can lead to a shortage of red blood cells and a condition called anemia. People with mild forms of anemia may feel tired or weak, but more severe forms of anemia can cause heart problems and even death.

A protein called mTOR forms part of a protein complex that helps alert the cells of many different organisms to the presence of nutrients. mTOR can add phosphate groups to ribosomes—the molecular machines that translate molecules of mRNA to build proteins. In 2012, researchers developed a technique called Phospho-Trap that can isolate these phosphorylated ribosomes from cells. Cells with an activated mTOR complex express more mTOR protein and in turn have more ribosomes that are modified. Examining the mRNA molecules associated with these ribosomes can reveal which proteins are produced in greater amounts in these cells.

Previous experiments using Phospho-Trap found the proteins that make up hemoglobin in unexpectedly high amounts in the mouse brain. Now, Knight et al.—and other researchers involved in the 2012 work—have established that the hemoglobin was not coming from the brain cells but from immature red blood cells circulating within the brain. These immature blood cells were found to have a highly active mTOR complex that promotes the production of hemoglobin and new blood cells.

Using genetic techniques in mice, Knight et al. found that the mTOR complex can cause anemia if it is underactive or overactive. Underactive mTOR complexes cause a type of anemia that produces small red blood cells and is usually triggered by a lack of iron. This made sense because mTOR is known to regulate both protein production and cell size. Boosting the activity of the mTOR complex leads to a type of anemia in which the cells are much larger than normal, and which is normally associated with inadequate amounts of folate and B12 vitamins.

When Knight et al. gave mice a drug that inhibits the mTOR protein, the mice developed anemia that resolved when the treatment stopped. However, mice that were given the mTOR inhibitor at the same time as a drug that destroys red blood cells, all died within days. Clinical trials are currently testing mTOR inhibitors as a possible cancer treatment; however, a common side effect of chemotherapy is that it stops new red blood cells being produced. Knight et al. suggest that the red blood cells of patients in these clinical trials must be closely monitored before deciding whether to continue the treatment further.

functions of these signaling complexes using cell-type-specific Cre drivers (*Chen et al., 2008*; *Gan et al., 2008*; *Gan and DePinho, 2009*; *Kalaitzidis et al., 2012*).

In a previous report, we developed a systematic approach to identify the cell types that show high levels of mTORC1 activity in vivo (*Knight et al., 2012*). This approach takes advantage of the fact that activation of mTORC1 leads to rapid phosphorylation of ribosomal protein S6 (rpS6) on five C-terminal serine residues (*Figure 1A*; Ser 235, 236, 240, 244, 247) (*Pende et al., 2004*; *Roux et al., 2007*). As rpS6 is a structural component of the ribosome, this phosphorylation introduces a 'tag' on the ribosomes from cells that have active mTORC1 signaling. We thus used phospho-specific antibodies to S6 to selectively immunoprecipitate these phosphorylated ribosomes from cells with high levels of TORC1 signaling in mouse brain homogenates, thereby enriching for the mRNA expressed in a subpopulation of activated cells (*Figure 1A*) (*Knight et al., 2012*).

Our previous report focused on the use of this approach to identify markers for activated neurons in the mouse brain. However, we also noted that genes encoding the protein subunits of hemoglobin were highly enriched in our pS6 immunoprecipitates. In this study, we report that these transcripts are derived from reticulocytes, immature red blood cells (RBCs), that we find have especially high levels of mTORC1 signaling. We further use a combination of pharmacologic, genetic, and nutritional perturbations to delineate a critical role for mTORC1 signaling in RBC development and the pathogenesis of

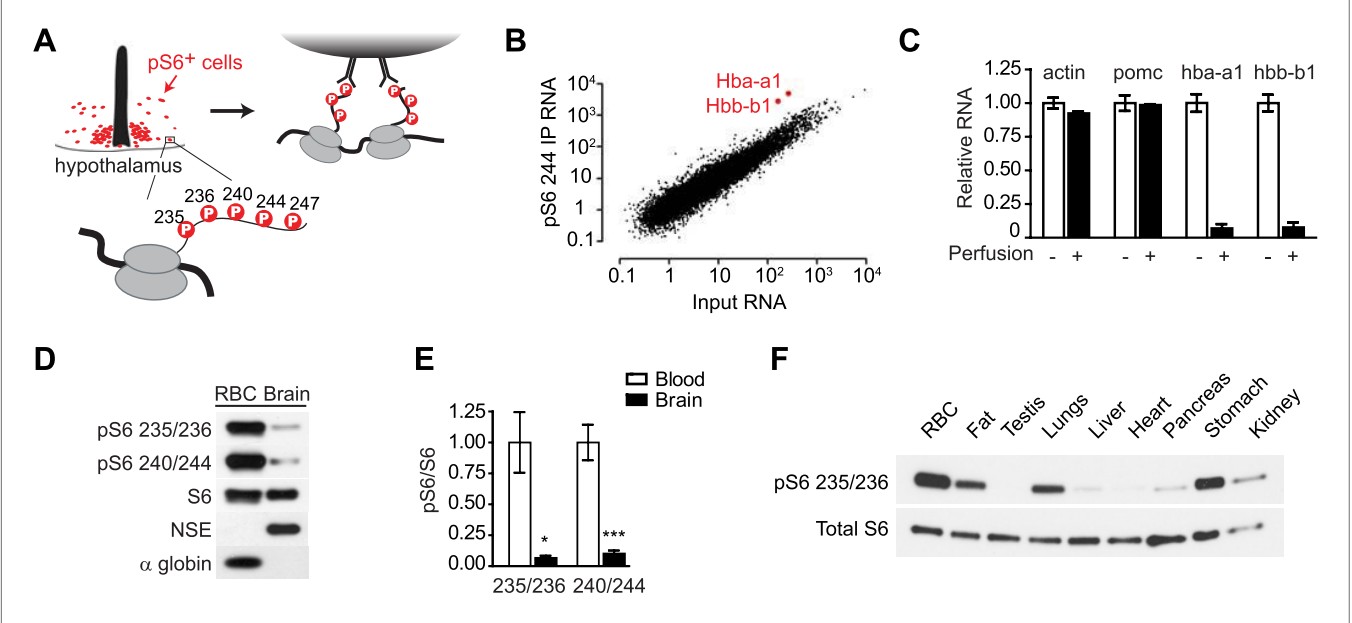

**Figure 1**. Reticulocytes have elevated mTORC1 signaling. (**A**) Diagram showing the phosphorylation sites on ribosomal protein S6 and strategy for immunoprecipitation of phosphorylated ribosomes. (**B**) Quantification by RNA-Seq of the abundance of each transcript in the pS6 244 immunoprecipitate (y-axis) vs the total hypothalamus (x-axis). Hba-a1 and hbb-b1 are enriched in the pS6 immunoprecipitate and labeled. (**C**) Total RNA was prepared from dissected hypothalami of mice that had been perfused with saline (black bars) or not perfused (white bars), and the relative abundance of each transcript was quantified by Taqman. Hba-a1 and hbb-b1 but not actin and pomc are depleted from the hypothalamus by perfusion. Values are normalized to rpl23. (**D**) Lysates from the mouse brain or RBCs were blotted for pS6 at the indicated sites. Neuron-specific enolase (NSE) and alpha globin are specific markers for the brain and RBCs, respectively. (**E**) Quantification of the relative phosphorylation of S6 at the indicated sites in the brain vs RBCs. Values are expressed as the ratio of pS6 to total S6. (**F**) Western blotting for pS6 235/236 in ribosomes purified from a range of mouse tissues. *p < 0.05. ***p < 0.001.

The following figure supplement is available for figure 1:

**Figure supplement 1**. α-globin is not expressed in a specific population of hypothalamic neurons.

anemia, suggesting that this pathway links the availability of iron to cell growth and hemoglobin synthesis during erythropoiesis.

## Results

### Reticulocytes have unusually high levels of pS6

We recently described a method for molecular profiling of activated neurons in the mouse brain (**Knight et al., 2012**). This approach takes advantage of the fact that ribosomal protein S6 is phosphorylated following neural activity (**Lenz and Avruch, 2005**; **Villanueva et al., 2009**; **Zeng et al., 2009**; **Valjent et al., 2011**; **Bertran-Gonzalez et al., 2012**). These phosphorylated ribosomes can then be immunoprecipitated from mouse brain homogenates, enriching for the mRNA expressed in a subpopulation of activated cells (**Figure 1A**).

During the course of these studies, we noticed that *Hba-a1* and *Hbb-b1* were highly enriched transcripts in pS6 immunoprecipitates from the mouse hypothalamus and other brain regions (**Figure 1B**). *Hba-a1* and *Hbb-b1* encode α- and β-globin, the protein subunits of hemoglobin. As hemoglobin is not highly expressed in the brain, the enrichment of these transcripts was unexpected and we set out to clarify their cellular origin.

We initially considered the possibility that hemoglobin might be expressed in a specific population of neurons that have high levels of pS6 at baseline. For example, VIP neurons of the suprachiasmatic nucleus (SCN) have high levels of pS6, and VIP mRNA is highly enriched in pS6 immunoprecipitates from the hypothalamus (**Figure 1—figure supplement 1**). However, consistent with the data from the Allen Brain Atlas, we were unable to detect specific α-globin expression in the SCN or any other

hypothalamic region by immunostaining or in situ hybridization. We thus considered the possibility that the globin RNA was not derived from a specific neural population but from another cell type (*figure 1—figure supplement 1*).

*Hba-a1* and *Hbb-b1* are most abundantly expressed in reticulocytes, immature RBCs that circulate in the blood. To test whether the *Hba-a1* and *Hbb-b1* transcripts originated from the circulating cells, we perfused mice with saline to remove blood from the tissue and then quantified the amount of globin mRNA remaining in hypothalamic extracts. Perfusion removed approximately 95% of *Hba-a1* and *Hbb-b1* mRNA from hypothalamus but had no effect on transcripts expressed in neurons or glia, such as *Actb* or *Pomc* (*Figure 1C*). These data show that the vast majority of *Hba-a1* and *Hbb-b1* mRNA in the brain originates from the circulating cells. To determine if *Hba-a1* and *Hbb-b1* were the only enriched erythroid transcripts in the blood, we scanned the RNAseq data for altered expression of other genes expressed in cells of the erythropoietic lineage. In contrast to transcripts for Hbb, we failed to find enrichment for erythroid catalase, carbonic anhydrase II, two cytoplasmic proteins, or sprectrin-a, spectrin-b, and ankyrin, which are membrane proteins.

The observation that the globin transcripts in our pS6 immunoprecipitates were derived from circulating cells suggested that reticulocytes were the source of this RNA and that reticulocytes might have unusually high levels of pS6. Furthermore, since pS6 is widely used as a marker for the activation of the mTORC1 pathway (*Meyuhas, 2008*), the data further suggested that reticulocytes might have particularly high levels of mTORC1 signaling. To test these possibilities, we first used western blotting to quantify the level of pS6 in the lysates from the brain and RBCs. Consistent with our ribosome profiling data, reticulocyte lysates had a much higher level of pS6 at both Ser 235/236 and Ser 240/244 compared to extracts from the brain as a whole (*Figure 1D,E*). We then extended this analysis by purifying the ribosomes from a panel of mouse tissues, including fat, testis, lungs, liver, heart, pancreas, stomach, and kidney, and then analyzed the level of pS6 in these tissues compared to blood. Remarkably, reticulocytes had the highest level of pS6 in any tissue we examined (*Figure 1F*). Thus, our ribosome profiling data from the brain revealed the unexpected finding that mTORC1 signaling is highly active in the RBC lineage.

## mTORC1 signaling in RBCs is regulated by iron

Reticulocytes are highly specialized for hemoglobin synthesis, and we wondered whether the elevated mTORC1 activity in this cell type might reflect its uniquely high demand for protein translation. The mTORC1 pathway plays a general role in linking nutrient availability to protein translation in diverse tissues (*Sengupta et al., 2010*), but relatively little is known about its function and regulation in RBCs. However, recent work has shown that mTORC1 can be regulated by the availability of iron in cell lines (*Ohyashiki et al., 2009*) and the brain (*Ndong et al., 2009*; *Fretham et al., 2013*), and, in turn, that mTORC1 activity can modulate downstream enzymes that control intracellular iron metabolism (*Bayeva et al., 2012*; *La et al., 2013*). These data raise the possibility that mTORC1 activity in RBCs might help coordinate the rate of translation with the availability of iron, although this had not been directly tested.

To test whether iron can regulate mTORC1 signaling in RBCs, wild-type mice were made iron deficient by placing them on a low iron diet combined with daily injections of the iron chelator deferoxamine. Complete blood counts were performed after 1 month and showed that the treated animals had developed microcytic anemia: the low iron cohort showed a reduction in RBC volume (58.7 vs 52.3 fL, p < 0.001 for reticulocytes; 48.3 vs 46.1 fL for mature RBC, p < 0.001) and RBC hemoglobin content (15.3 vs 13.2 pg for reticulocytes, p < 0.001; 14.0 vs 13.4 pg for mature RBC, p < 0.01) relative to control animals (*Figure 2A,B*). To assess the level of mTORC1 signaling in these cells, we prepared lysates from RBCs from animals in each cohort and performed western blots for pS6. Iron deficiency greatly reduced the level of pS6 in RBCs at both Ser 235/236 as well as Ser 240/244, the latter of which is a specific marker for mTORC1 activity (*Roux et al., 2007*; *Meyuhas, 2008*) (*Figure 2C,D*). Consistent with previous data, we observed a similar inhibition of mTORC1 signaling when we treated the erythroleukemia cell line K562 with iron chelators in vitro (*Figure 2E*), (*Ohyashiki et al., 2009*). Thus iron deficiency results in a marked reduction in mTORC1 signaling in RBCs in vitro and in vivo.

## mTORC1 activation in RBCs produces macrocytic anemia

The regulation of mTORC1 signaling by iron suggests a possible mechanism for linking nutrient status, in this case iron availability, to RBC size. To explore the role of mTORC1 in RBC growth and

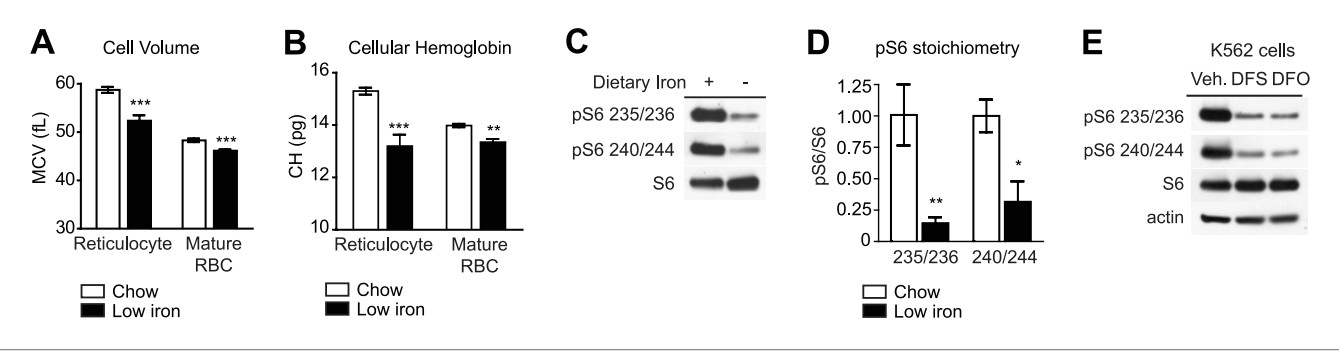

**Figure 2**. mTORC1 signaling in RBCs is regulated by iron. (**A**) Mean cell volume (MCV) of reticulocytes and mature RBCs in mice that were challenged with a low iron diet (black bars) or maintained on a chow diet (white bars) for 1 month. (**B**) Cellular hemoglobin of reticulocytes and mature RBCs in mice that were challenged with a low iron diet (black bars) or maintained on a chow diet (white bars) for 1 month. (**C**) Western blot for pS6 from RBCs in mice that that were challenged with a low iron diet (black bars) or maintained on a chow diet (white bars) for one month. (**D**) Quantification of the relative phosphorylation of S6 at the indicated sites in the RBCs from mice on a low iron (black bars) or chow (white bars) diet. Values are expressed as the ratio of pS6 to total S6. (**E**) K562 cells were treated with the iron chelator deferoxamine (DFO), deferasirox (DFS), or vehicle (0.05% DMSO) for 24 hr, and then lysed and levels of pS6 analyzed by western blotting. *p < 0.05. ***p < 0.001.

hemoglobin synthesis, we first used a genetic approach to either increase or decrease mTORC1 signaling selectively in hematopoietic cells. mTORC1 is under tonic inhibition by the Tsc1/Tsc2 complex, and *Tsc1* deletion results in constitutively increased mTORC1 signaling. Previous work has shown that inducible deletion of *Tsc1* in adult animals using either *Mx1^Cre* or *Gt(ROSA)26Sor^Cre*-ERT2 promotes rapid cycling of the hematopoietic stem cells (HSC) and is associated with decreased hematopoiesis due to HSC exhaustion (***Chen et al., 2008***; ***Gan et al., 2008***; ***Gan and DePinho, 2009***). However, *Mx1* and *Gt(ROSA)26Sor* are expressed in multiple tissues and do not provide information on the effects of cell autonomous *Tsc1* deletion in the RBC lineage. We thus assessed the effect of mTORC1 activation in hematopoietic cells by using *Vav1^Cre* mice to delete a floxed allele of *Tsc1* (***Kwiatkowski et al., 2002***). Vav-1 is a guanosine nucleotide exchange factor that is exclusively expressed in the hematopoietic lineage (***Zhang et al., 1994***). We selected a *Vav^Cre* driver because it has been reported to catalyze highly efficient excision of loxP flanked sequences in erythrocytes and in previous studies enabled analyses of several different loss of function phenotypes in the RBC lineage (***Pan et al., 2007***; ***Gan et al., 2010***; ***Mortensen et al., 2010***; ***Schultze et al., 2012***). In contrast, several other *Cre* lines that use different erythrocyte or hematopoietic-specific promoters have been shown to often produce variegated effects with incomplete recombination in RBCs (***Peterson et al., 2004***; ***Mortensen et al., 2010***).

*Vav1^Cre Tsc1^fl/fl* animals (hereafter called TSC-KO) were born at the expected ratio but were smaller than their littermates and had increased mortality in the neonatal period. However, animals that survived to 8 weeks of age had a normal appearance. Western blotting revealed that RBCs from TSC-KO mice had markedly increased levels of pS6 relative to littermate controls, confirming that these cells have constitutive activation of mTORC1 signaling (***Figure 3A***).

Analysis of peripheral blood from TSC-KO mice revealed a constellation of changes consistent with macrocytic anemia (***Figure 3***). Compared to littermate controls, TSC-KO mice had larger reticulocytes (66.7 vs 58.6 fL, p < 0.0001) and larger mature RBCs (54.5 vs 48.1 fL, p < 0.0001) and the larger RBCs from TSC-KO mice likewise had significantly more hemoglobin (17.4 vs 15.1 pg in reticulocytes; 16.2 vs 13.9 pg in mature RBCs, p < 0.0001). In addition, the percentage of cells with elevated hemoglobin was dramatically increased in both young and old RBCs (29.2 vs 5.00% in reticulocytes; 13.5 vs 0.58% in mature RBCs, p < 0.0001).

Despite this increase in RBC size and hemoglobin content, TSC-KO mice were anemic compared to littermate controls (39.4 vs 45.7% hematocrit, p < 0.0001; ***Figure 3C***). The reduced hematocrit of TSC-KO mice resulted from a reduction in the RBC number (7.58 vs 9.89 × 10^9 cells/mL, p < 0.0001), such that the total amount of hemoglobin in the peripheral blood was reduced (1.12 vs 1.28 g/dL, p < 0.01). Thus constitutive activation of mTORC1 leads to a macrocytic anemia with an increase in hemoglobin per cell (***Chen et al., 2008***; ***Gan et al., 2008***; ***Gan and DePinho, 2009***).

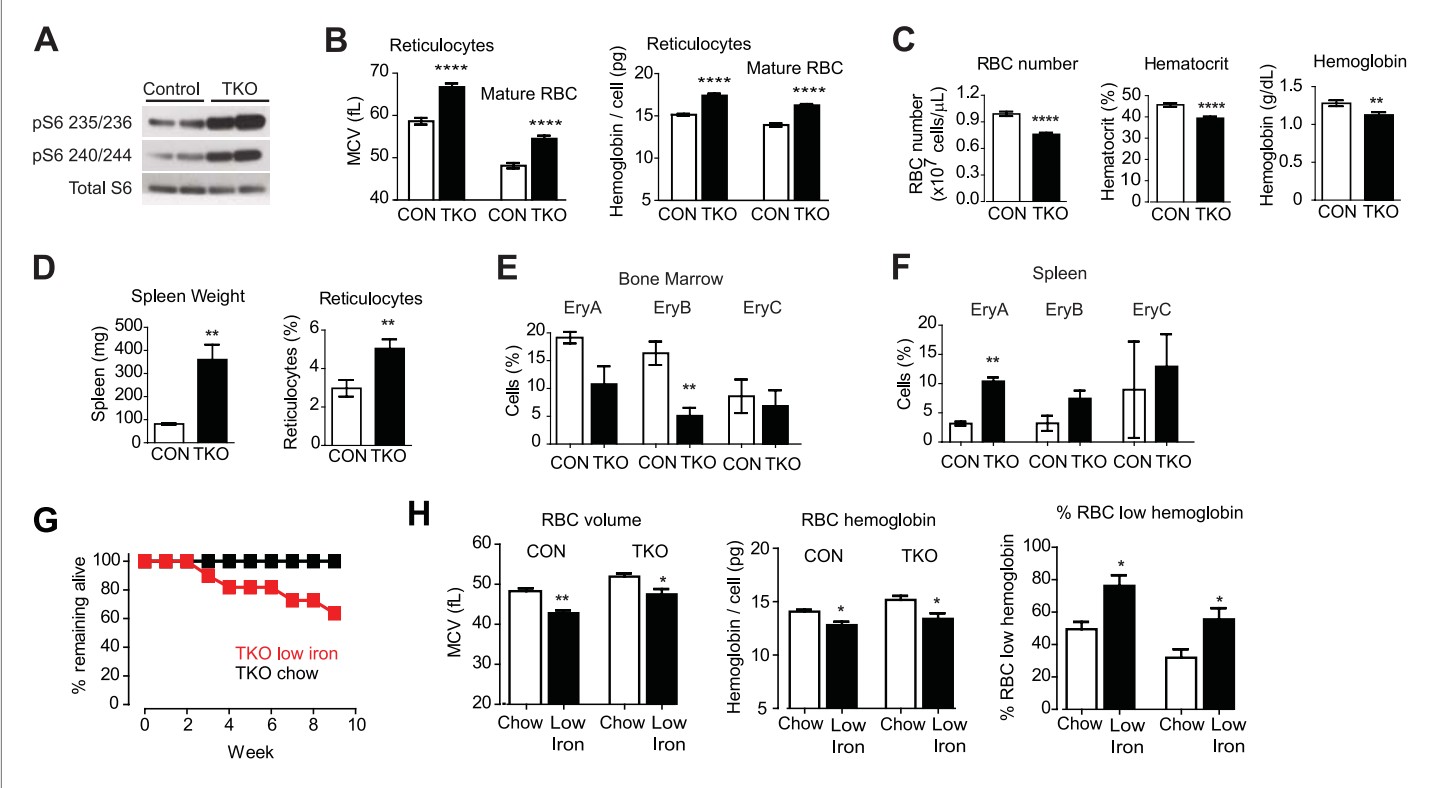

**Figure 3**. Tsc1 deletion in hematopoietic cells results in hyperchomic macrocytic anemia. (**A**) Western blots for pS6 at Ser 235/236 and Ser 240/244 in Vav$^{Cre}$Tsc1$^{fl/fl}$ mice (TSC-KO) or littermate controls. (**B**) Mean cell volume (MCV) and hemoglobin content per cell of reticulocytes and mature RBCs from control (white bars) and TSC-KO (black bars) mice. (**C**) RBC number, hematocrit, and total blood hemoglobin from control (white bars) and TSC-KO (black bars) mice. (**D**) Spleen weight and reticulocyte percentage from control (white bars) and TSC-KO (black bars) mice. (**E**) Percentage of bone marrow cells corresponding to each erythrocyte progenitor subtype from control (white bars) and TSC-KO (black bars) mice. Values were determined by flow cytometry for Ter119 and CD71 as described in methods. (**F**) Percentage of splenocytes corresponding to each erythrocyte progenitor subtype from control (white bars) and TSC-KO (black bars) mice. (**G**) Percentage of TSC-KO mice remaining alive at the indicated times after transition to low iron diet (red) or control diet (black). No mortality was observed in littermate controls on either diet. (**H**) Quantification of mean cell volume, hemoglobin content per cell, and percentage of cells with low hemoglobin, in control and TSC-KO mice exposed to a control (white bars) or low iron (black bars) diet. *p < 0.05, **p < 0.01, ****p < 0.0001.

We also noticed that TSC-KO mice had a higher percentage of circulating reticulocytes (5.04 vs 2.97%, p < 0.01) with more than threefold increase in the size of their spleens (360 vs 80.5 mg, p < 0.01) relative to littermate controls (**Figure 3D**). These changes are often characteristic of stress erythropoiesis, suggesting erythropoiesis in the spleen of TSC-KO mice is activated to compensate for reduced RBC production by the bone marrow. To confirm this, we isolated splenocytes and bone marrow cells (BMCs) from TSC-KO and control mice and used flow cytometry to quantify the abundance of RBC progenitors in these two compartments. Consistent with this possibility, TSC-KO mice showed a decrease in the number of erythroid progenitors in the bone marrow (EryB: 5.08 vs 16.3, p < 0.01), and a reciprocal increase in the number of erythroid progenitors in the spleen (EryA: 3.15 vs 10.4%, p < 0.01) (**Figure 3E**). Thus, we find that constitutive activation of mTORC1 is sufficient to recapitulate a constellation of features that characterize macrocytic anemia, including an increase in RBC size and hemoglobin content coupled to a decrease in RBC counts and hematocrit.

## Iron deficiency decreases survival of TSC-KO mice

RBCs must balance the rate of globin synthesis with the availability of iron, so that free globin peptides do not aggregate in the absence of heme. An excess of free globin decreases red blood cell survival in a number of conditions including thalassemia. Thus, chronic iron deficiency triggers an adaptive response in which hemoglobin synthesis and RBC size are reduced, resulting in a microcytic

hypochromic anemia in which the low heme levels are balanced by lower levels of globin synthesis. During iron deficiency the activity of heme-regulated eIF2alpha kinase (HRI), a protein kinase that is selectively expressed in RBCs is increased and acts to reduce hemoglobin synthesis. This kinase is phosphorylated and inhibited by free heme and is disinhibited with iron depletion (*Chen, 2007*). HRI knockout mice do not reduce globin synthesis after iron deficiency indicating that activation of this kinase plays an important role in the adaptive response to dietary iron deficiency (*Han et al., 2001*).

However, since mTORC1 is inhibited by iron deficiency (*Figure 2*), this raised the question of whether the constitutive mTORC1 activation could also exacerbate the response to iron deficiency. To test this we challenged two cohorts of TSC-KO mice as well as littermate controls with either a low iron or control diet for two months and then measured the effect on RBC growth and proliferation. Similar to the effects of an HRI deficiency on the response to iron deficiency, TSC-KO mice became visibly moribund and showed progressive mortality beginning at 3 weeks on an iron deficient diet, such that only 64% of the TSC-KO mice survived the 2-month experiment (*Figure 3G*). By contrast, littermate controls were more normal in appearance and had a higher survival. Thus chronic mTORC1 activation in RBCs is maladaptive in the context of iron deficiency.

Complete blood counts after 2 months revealed that, similar to controls, TSC-KO mice showed a reduction in the RBC volume and hemoglobin content on the low iron diet. (*Figure 3H*). The iron deficient TSC-KO mice also displayed splenomegaly and reticulocytosis. We also noted premature mortality in TSC-KO mice whether or not they were on a low iron diet. Autopsies of two TSC-KO mice, one on chow and another on an iron deficient diet at 6 months of age were performed. Both mice had extramedullary hematopoiesis with multifocal myeloid hyperplasia which was described as moderate. Consistent with the greater mortality of animals on low iron diet, the iron deficient TSC-KO mouse showed increased number of phagocytosed red blood cells in multiple organs compared to the chow-fed TSC-KO animal. These data are consistent with the possibility that constitutive activation of the mTOR pathway in the setting of iron deficiency, which was further exacerbated by phlebotomy for blood collection, decreased RBC viability and increased mortality at younger ages. (*Kazemi et al., 2007*)

All of the TSC-KO mice died prematurely by ~6 months of age and both animals also developed histiocytic sarcomas which showed diffuse infiltration of neoplastic round cells in thymus, lung, liver, kidneys, spleen, bone marrow, meninges, uterus, and lymph nodes. Note, a previous report showed that animals with increased mTOR signaling as a result of a *Pten* mutation in HSCs using *Mx1^Cre* mice also developed histiocytic sarcomas (*Lee et al., 2010*).

## mTORC1 ablation produces hypochromic microcytic anemia and perinatal lethality

Since mTORC1 activity is increased by iron and constitutive mTORC1 activation results in macrocytic anemia, we next considered the possibility that mTORC1 inhibition would cause microcytic anemia. Germline deletion of either mTOR or the essential mTORC1 component Raptor results in embryos that die shortly after implantation (*Guertin et al., 2006*), which has complicated analysis of the requirement for mTORC1 signaling in erythropoiesis. Experiments using inducible deletion of mTORC1 subunits in the hematopoietic cells have yielded variable results. For example, an inducible deletion of *Rptor* in either *Mx1^Cre* or *ERT2^Cre* animals was reported to be well tolerated in adult animals (*Hoshii et al., 2012*; *Kalaitzidis et al., 2012*). By contrast, inducible deletion of the mTOR kinase in adult mice has recently been reported to result in severe anemia and lethality within 17 days (*Guo et al., 2013*). The more dramatic phenotype following mTOR deletion may reflect the role of other mTOR complexes in erythropoiesis or reflect differences in recombination efficiency and selectivity between these experiments. To directly address the role of mTOR during the development of the hematopoietic system, we mated *Vav1^Cre* to mice with a floxed allele of *Rptor* (*de Boer et al., 2003*).

Initial crosses between *Vav1^Cre Rptor^{fl/+}* and *Rptor^{fl/fl}* animals failed to yield any viable *Vav1^Cre Rptor^{fl/fl}* (hereafter called RAP-KO) offspring (>100 offspring genotyped from >20 litters). The *Vav1^Cre* transgene is expressed beginning at E13, which corresponds to the onset of definitive erythropoiesis in the fetal liver. We therefore dissected and analyzed embryos from these crosses between E13 and E17.5 and found that embryos with the *Vav1^Cre Rptor^{fl/fl}* genotype were present at the expected ratio and morphologically normal. However, RAP-KO embryos were strikingly pale compared to their littermates (*Figure 4A*). Pallor during development is characteristic of a diverse set of mouse mutants that have a defect in definitive erythropoiesis. Mutations that cause a complete block of erythropoiesis, such as loss of erythropoietin or its receptor, lead to embryonic lethality at approximately E13 (*Wu et al.,*

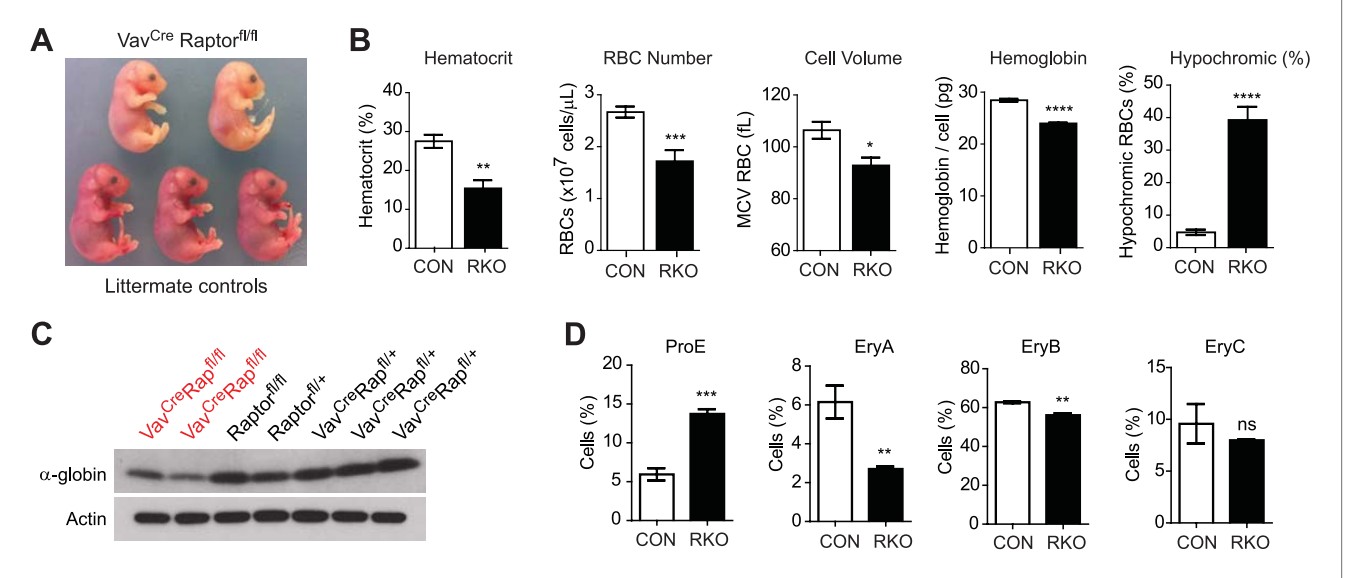

**Figure 4**. Raptor deletion in hematopoietic cells results in hypochromic microcytic anemia and perinatal lethality. (**A**) Embryos dissected at E16.5 illustrating the pallor of Vav$^{Cre}$Raptor$^{fl/fl}$ (RAP-KO) mice relative to littermate controls. (**B**) Complete blood counts from the fetal liver of E17 RAP-KO embryos and littermate controls. (**C**) Western blotting for the level of α-globin and actin in RAP-KO mice or littermate controls. (**D**) Quantification of the percentage of fetal liver cells corresponding to various erythroid progenitors at E16.5 by flow cytometry. *p < 0.05, **p < 0.01, ***p < 0.001, ****p < 0.0001.

The following figure supplement is available for figure 4:

**Figure supplement 1**. Newborn RAP-KO pups are extremely pale.

---

1995). In contrast, we found that many RAP-KO embryos continued to develop throughout gestation. Monitoring of the delivery from pregnant females from these crosses led to the identification of several live-born RAP-KO pups, but, similar to the embryos, these animals were extremely pale and survived less than 1 day (*Figure 4—figure supplement 1*).

*Vav$^{Cre}$* is expressed in all cells of the hematopoietic lineage. Therefore, the perinatal lethality of RAP-KO embryos could in principle result from *Rptor* deletion in other cell types such as lymphocytes. To exclude this possibility, we knocked out *Rptor* specifically in lymphoid cells using an *hCD2$^{Cre}$* driver. In contrast to RAP-KO mice, *hCD2$^{Cre}$Rptor$^{fl/fl}$* mice that have *Rptor* deleted selectively in the B and T cells (*de Boer et al., 2003*) had normal birth-rates and survival, indicating that Raptor loss in the lymphocytes is unlikely to contribute to the perinatal lethality of RAP-KO animals. Thus there were no other major defects in any of the other hematopoietic compartments that might have contributed to their early lethality; lymphocyte percentages were comparable in the RAP-KO with their littermates, and the total white cell count was higher (p < 0.01) in the RAP-KO. There were no significant differences in the number of basophils, platelets, and monocytes. These data show that the abnormalities in the hematopoietic system of mice with Vav-cre specific inactivation of mTORC1 are limited to erythroid cells.

Consistent with their visual pallor, hematological analysis of fetal livers from RAP-KO embryos revealed a broad impairment in erythropoiesis (*Figure 4B*). RAP-KO embryos were anemic (Hct 15.4 vs 27.5%, p < 0.01) with reduced RBC numbers (1.72 vs 2.67 × 10^9 cells/mL, p < 0.001). Analysis of specific erythroid progenitors by flow cytometry (*Figure 4D*) revealed that RAP-KO embryos have an increased abundance of the earliest hematopoietic progenitors (ProE, 5.95 vs 13.7%, p < 0.001) and a decrease in the abundance of more mature populations (EryC, 6.15 vs 2.70%, p < 0.01) relative to littermates. This shift in the distribution of fetal liver progenitors is similar to what has been observed in previous studies of the bone marrow of adult mice following inducible deletion of the mTOR kinase, though in these studies the hematopoietic phenotype was attributed to increased apoptosis of bone marrow RBCs (*Guo et al., 2013*).

We also found that, similar to the effect of iron deficiency, RBCs from RAP-KO embryos were smaller than those from littermate controls (92.8 vs 106 fL, p < 0.05) and had less total hemoglobin

(24.0 vs 28.5 pg/cell, p < 0.0001) (*Figure 4B*). The defect in hemoglobin synthesis was particularly severe, as there was an eightfold increase in the percentage of cells that were hypochromic in RAP-KO embyros (39.3 vs 4.71, p < 0.0001). We confirmed this reduction in hemoglobin expression biochemically by western blotting, which showed that RAP-KO embyros had reduced α-globin expression compared to the littermate controls (*Figure 4C*). Thus genetic ablation of mTORC1 produces a severe microcytic hypochromic anemia that results in perinatal lethality. These data are consistent with the possibility that a reduction of mTORC1 plays a role in the development of the hypochromic microcytic anemia of iron deficiency. However, because the *Rptor* knockout was lethal during development, we were unable to use genetics to assess this possibility in adult animals. As an alternative, we employed a pharmacologic approach to inhibit mTORC1 in adult mice.

## Pharmacologic mTOR inhibition acutely impairs erythropoiesis in adults

In previous studies, treatment of adult animals with the mTORC1 inhibitor rapamycin was shown to have minimal effects on erythropoiesis. For example, chronic rapamycin treatment of rats results in microcytosis without the development of anemia (*Diekmann et al., 2012*). Likewise, the phenotype of mice following an inducible *Cre*-mediated deletion of *Rptor* has led to the conclusion that mTORC1 may be dispensable for the regulation of hematopoiesis in adults (*Hoshii et al., 2012*; *Kalaitzidis et al., 2012*) (but see also *Guo et al., 2013*).

To examine the acute requirement for mTOR signaling in adult erythropoiesis, we treated normal mice for 3 days with either rapamycin or MLN0128, a selective, ATP competitive mTOR inhibitor that is being clinically tested as a treatment for several types of human cancer (*Hsieh et al., 2012*; *Infante et al., 2012*). We assayed the effect of these drugs on RBCs in the peripheral blood and bone marrow. Consistent with the previous reports, we found that rapamycin had only a modest

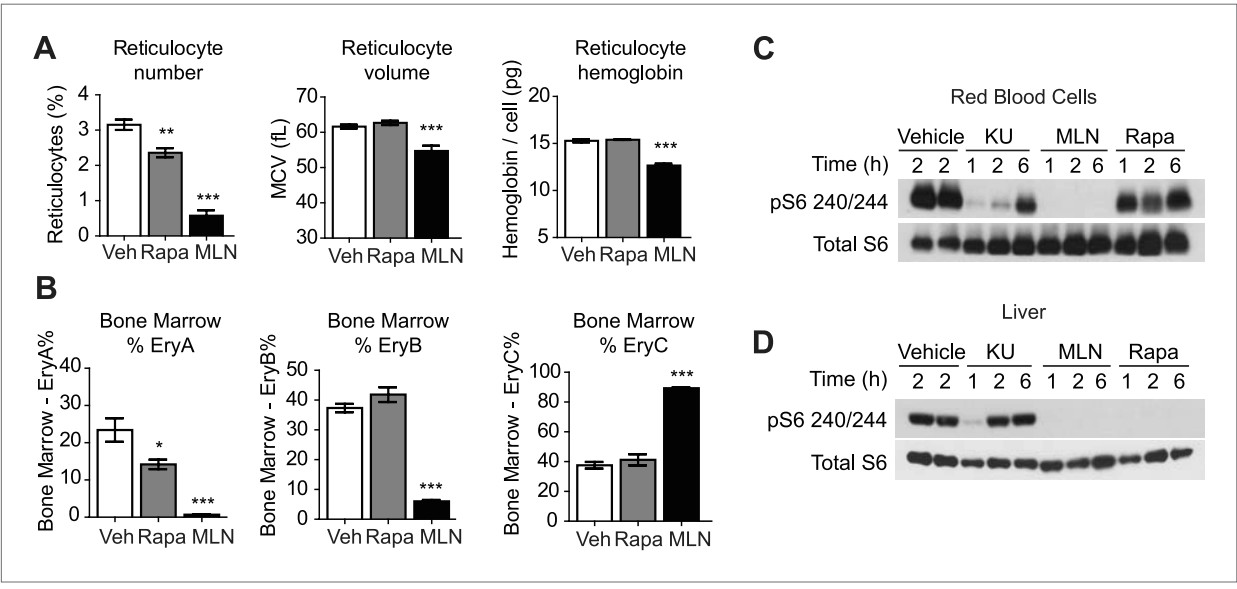

**Figure 5**. ATP competitive mTOR inhibitors but not rapamycin block S6K signaling, growth, and proliferation of RBCs. (**A**) Reticulocyte percentage, mean cell volume, and hemoglobin content per cell in mice treated with MLN0128, rapamycin, or vehicle for 3 days. (**B**) Quantification by flow cytometry of the percentage of bone marrow cells corresponding to various erythroid progenitors following treatment with MLN0128, rapamycin, or vehicle for 3 days. (**C**) Western blotting for the level of pS6 240/244 and total S6 in RBCs isolated from mice treated with vehicle, KU-0063794 (10 mg/kg), MLN0128 (2 mg/kg), or rapamycin (10 mg/kg) for 1, 2, or 6 hr. (**D**) Western blotting for the level of pS6 240/244 and total S6 from the livers of the same animals as in panel (**C**). *p < 0.05, **p < 0.01, ***p < 0.001.
The following figure supplement is available for figure 5:

**Figure supplement 1**. RBCs have intrinsically reduced sensitivity to rapamycin.

impact on adult erythropoiesis, with a reduction in the percentage of circulating reticulocytes (3.15 vs 2.36%, p < 0.01) but no change in reticulocyte size or hemoglobin content (*Figure 5A*).

In contrast, acute MLN0128 treatment had profound effects on RBC growth and proliferation (*Figure 5A*). The percentage of circulating reticulocytes was reduced by more than 75% (3.15 vs 0.58%, p < 0.001) and the remaining reticulocytes were smaller (61.6 vs 54.7 fL, p < 0.001) and had less hemoglobin per cell (15.3 vs 12.7 pg, p < 0.001). The reduction in reticulocyte number was paralleled by a decrease in the proliferation of erythroid progenitors in the bone marrow (*Figure 5B*). MLN0128 dramatically reduced the abundance of the early erythroid progenitors EryA (23.4% vs 0.7%, p < 0.001) and EryB (37.3% vs 6.0%, p < 0.001). By contrast, rapamycin caused more modest reduction in only the EryA population (23.4% vs 14.2%, p < 0.05). Thus acute mTOR inhibition by MLN0128, but not rapamycin, strongly blocked RBC growth, hemoglobin synthesis, and the maturation of erythroid progenitors in the bone marrow of treated mice.

MLN0128 fully inhibits both mTORC1 and mTORC2 (*Hsieh et al., 2012*), whereas rapamycin blocks only a subset of signaling pathways that are activated by mTORC1 (*Feldman et al., 2009*). This raised the possibility that the differences between the effects of these two drugs could be a result of inhibition of mTORC2 by MLN0128 but not rapamycin. Alternatively, it is possible that mTORC1 in RBCs was not inhibited by rapamycin to the same extent as it was by MLN0128. In a prior report, a genetic deletion of the essential mTORC2 component rictor had little effect on hematopoiesis, suggesting that TORC2 does not play a major role (*Magee et al., 2012*). For this reason, we compared first the effects of MLN0128 and rapamycin on mTORC1 signaling. Mice were treated with either rapamycin, MLN0128, or a third, structurally unrelated ATP competitive mTOR inhibitor (KU-0063794) (*Garcia-Martinez et al., 2009*), and the level of pS6 in RBCs was analyzed by western blotting. MLN0128 and KU-0063794 both potently inhibited S6 phosphorylation in RBCs (*Figure 5C*). The effect of these drugs was similar at 1 and 2 hr, although MLN0128 showed more durable inhibition at later time points (*Figure 5C*). By contrast, we unexpectedly found that rapamycin only weakly inhibited S6 phosphorylation in RBCs (*Figure 5C*) and that this rapamycin resistance could not be overcome even by chronic rapamycin treatment (daily for 4 weeks) at high doses (10 mg/kg). This effect was specific to RBCs, as rapamycin potently and durably blocked S6K signaling in other tissues from the same animals (such as liver, *Figure 5D*). Because S6 is not a direct target of mTORC1, we also assessed the phosphorylation state of 4E-BP1 in animals treated with MLN0128 (*Figure 6B*). These data showed that phosphorylation of 4E-BP1 was markedly decreased in treated animals. Furthermore, the extent of this reduction was highly correlated with the reduction in the level of phosphorylation of S6 (*Figure 6B*).

The reduced effect of rapamycin was also cell autonomous, as RBCs cultured in vitro were sensitive to ATP competitive mTOR inhibitors but not rapamycin (*Figure 5—figure supplement 1*). Thus, mTORC1 signaling through the S6K pathway is selectively resistant to rapamycin in RBCs. While certain mTORC1 substrates such as 4E-BP1 are known to display variable rapamycin sensitivity (*Feldman et al., 2009*), rapamycin resistant S6K signaling has to our knowledge never been described (*Choo and Blenis, 2009*). This unexpected reduction in the sensitivity of RBCs to rapamycin is consistent with our observation that ATP competitive mTOR inhibitors cause a more pronounced inhibition of RBC growth and proliferation than rapamycin.

## mTORC1 signaling is required for survival under conditions of stress erythropoiesis

The effects of MLN0128 on RBC growth and proliferation were greatly diminished after 4 weeks of treatment, at which point the RBC counts and hemoglobin levels of treated mice were nearly normal. This result suggests that there is tachyphylaxis or some other homeostatic response that can compensate for the chronic but not acute mTOR inhibition in RBCs. As acute erythropoiesis is particularly critical under conditions of erythropoietic stress, we next assayed the effect of mTOR inhibitors following drug-induced hemolysis (*Paulson et al., 2011*). Mice were treated with phenylhydrazine on days 0 and 2 along with daily injections of either an mTOR inhibitor or vehicle. Phenylhydrazine causes irreversible oxidative damage to RBCs (*Figure 6C*, right) and triggers acute hematopoiesis with a shift in erythropoiesis from the bone marrow to the spleen (*Figure 6E*), the proliferation of splenic progenitors (*Figure 6D*), and the release of stress reticulocytes into the circulation and that rapidly repopulate the hematopoietic system (*Figure 6C*, center).

Vehicle-treated mice showed a normal response to phenylhydrazine-induced hemolysis and recovered normally within 1 week (*Figure 6A*). By contrast, mice treated with MLN0128 were visibly moribund by

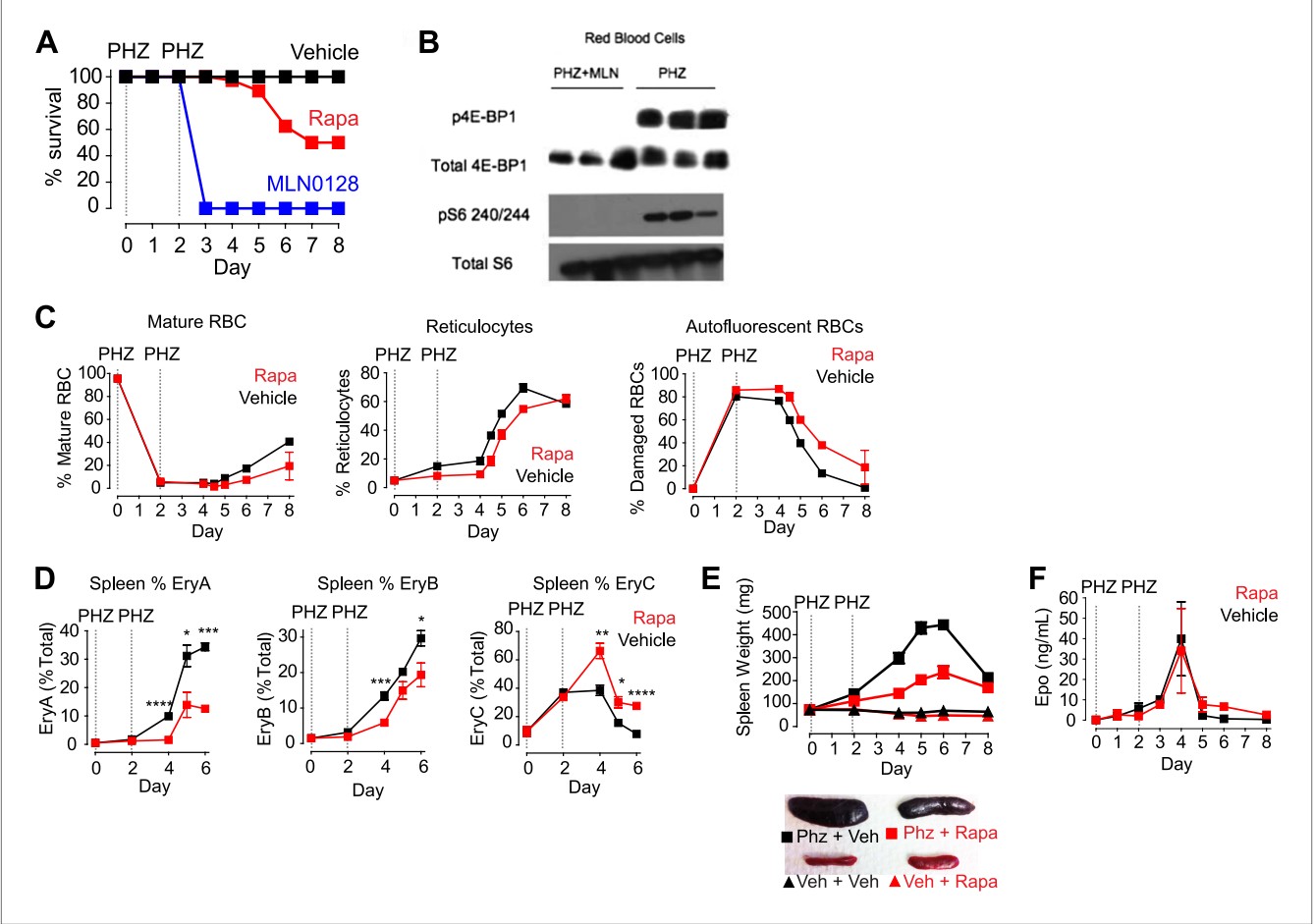

**Figure 6**. mTORC1 signaling is required for induction of stress erythropoiesis. (**A**) Survival of mice treated with MLN0128 (1 mg/kg), rapamycin (10 mg/kg) or vehicle following challenge with phenylhydrazine injection (50 mg/kg) on days 0 and 2. No lethality was observed by injection of either drug or vehicle alone. (**B**) Western Blotting for the level of p4E-BP1 and 4E-BP1 and pS6 240/244 and total S6 in phenylhydrazine and MLN0128-treated mice. (**C**) Time course of changes in peripheral blood populations following phenylhydrazine challenge of mice treated daily with vehicle or rapamycin. Analysis performed by flow cytometry using Retic-Count as desribed in methods. Autoflourescent RBCs represent damaged RBCs that presumably contain Heinz bodies composed of damaged hemoglobin. (**D**) Time course of changes in erythroid progenitors in the spleen determined by flow cytometry following phenylhydrazine challenge of mice treated daily with vehicle or rapamycin. (**E**) Time course of spleen weight in mice challenged with phenylhydrazine (square) or vehicle (triangle) and then treated daily with rapamycin (red) or vehicle (black). Images of representative spleens from each group are shown below. (**F**) Concentration of erythropoietin in the peripheral blood from mice challenged with phenylhydrazine and then treated with rapamycin (red) or vehicle (black) daily. *p < 0.05, **p < 0.01, ***p < 0.001, ****p < 0.0001.

day 2, with reduced mobility, weight loss, and a reduction in body temperature. By day 3 all animals had died or had to be euthanized for animal welfare reasons (*Figure 6A*). The dramatic mortality of animals treated with MLN0128 indicates that mTOR signaling is essential for the acute response to hemolytic stress. Western blots were performed using protein from the RBCs of phenylhydrazine-treated mice with and without MLN0128 administration. These data showed high levels of phosphorylation of p4E-BP1, a direct target of mTORC1 and pS6, an indirect mTorc1 target, in mice treated with phenylhydrazine alone. However, this phosphorylation was completely absent in phenylhydrazine-treated mice that also received MLN0128. (*Figure 6B*).

As we previously found that rapamycin only partially inhibited mTORC1 signaling in RBCs, we also tested the effect of this drug in the same assay. Approximately 50% of rapamycin-treated animals survived the 8-day experiment, and the onset of mortality occurred several days later than the animals treated with MLN0128 (*Figure 6A*). Rapamycin treatment also delayed the release of stress reticulocytes into the blood (*Figure 6C*) and reduced by twofold the increase in spleen size that occurred

following phenylhydrazine (*Figure 6E*). These results indicate that the partial mTORC1 inhibition by rapamycin delays but does not block the induction of stress erythropoiesis.

To characterize rapamycin's effects on stress erythropoiesis in more detail, we performed a time course analysis of erythroid progenitors in the spleen by flow cytometry in animals treated with phenylhydrazine with and without rapamycin. Early erythroid progenitors (EryA and EryB) were present at very low levels in the spleen on day 0 but were rapidly mobilized by phenylhydrazine treatment in control animals, reaching more than 30% of total splenocytes by day 6 (*Figure 6D*, black). This proliferative response was significantly reduced at all time points tested by rapamycin treatment (*Figure 6D*, red). By contrast, rapamycin treatment had no effect on erythropoietin levels, which, similar to control mice, increased approximately 400 fold following phenylhydrazine treatment. This suggests that the defect induced by rapamycin results from mTORC1 inhibition in the red cell lineage (*Figure 6F*).

Taken together, these data reveal a critical role for mTORC1 signaling in mediating the rapid growth and proliferation of erythroid progenitors during stress erythropoiesis. ATP competitive inhibitors block this response much more potently than rapamycin, resulting in a lethal failure to adapt to hemolytic stress. As MLN0128 and other ATP competitive mTOR inhibitors are currently being evaluated in clinical trials, these results suggest that further studies will be necessary to assess the effect of these agents under conditions that trigger stress erythropoiesis, such as hypoxia, blood loss, or drug-induced hemolysis.

## Discussion

The mTORC1 pathway coordinates protein translation with nutrient availability in organisms ranging from yeast to humans, and the physiologic functions of mTORC1 signaling in specific mammalian tissues have begun to be elucidated. Within the hematopoietic system, an extensive literature has investigated the role of mTORC1 signaling in leukocyte development and function (*Gan et al., 2008*; *Gan and DePinho, 2009*; *Janes et al., 2010*; *Hoshii et al., 2012*; *Kalaitzidis et al., 2012*; *Magee et al., 2012*; *Guo et al., 2013*), including its role in hematopoietic stem cells (HSCs) and rapamycin-mediated immunosuppression. By contrast, much less is known about the role of the mTORC1 pathway in erythrocytes.

Our attention was drawn to the potential significance of mTORC1 signaling in RBCs by screening for cell types in the adult brain that have elevated levels of pS6, a ribosomal protein that is a downstream target of mTORC1 activation (*Knight et al., 2012*). In that study, we used immunoprecipitation of polysomes with an anti-pS6 antibody to enrich for mRNAs from cells that had activated mTORC1 signaling. This study revealed an unexpected enrichment of globin transcripts in the precipitated polysomes. In this study, we report that circulating reticulocytes are the source of these enriched globin transcripts and that these cells have an especially high level of S6 phosphorylation. Since mTOR is known to link changes in nutrient availability to changes in translation and cell growth, these data suggested that mTOR could play an important role in fine tuning the physiologic state of RBCs in response to alterations in the availability of critical nutrients such as iron. We found that mTORC1 in RBCs is regulated by iron and that iron deficiency results in decreased mTORC1 activity in RBCs in vivo. We further showed that activation of mTORC1 by deletion of Tsc1 results in severe macrocytic anemia, while genetic inhibition of mTORC1 results in a lethal microcytic anemia. Thus bidirectional modulation of mTORC1 signaling is sufficient to cause either a macrocytic and microcytic anemia, both of which can be caused by nutrient deficiency. We further showed that pharmacologic inhibition of mTOR by ATP competitive inhibitors and to a lesser extent rapamycin resulted in an acute blockade of erythropoiesis in adults and that mTORC1 signaling is critically required for survival, following hemolytic stress. Finally, we identified rapamycin resistant S6K signaling in RBCs as a biochemical correlate of the differential effects of these drugs on RBC function. Thus, these studies delineate a critical role for mTORC1 signaling in RBC growth, development, and proliferation, and define how this pathway is selectively modulated by distinct pharmacologic, genetic, or nutritional perturbations leading to anemia.

While few studies have directly examined the role of mTORC1 signaling in RBCs, there is some evidence that dysregulation of protein translation downstream of mTORC1 may play a role in the development of anemia. The hereditary anemias, Diamond-Blackfan and 5q-syndrome, are caused by haploinsufficiency of ribosomal proteins S14 and S19, and recent data show that these anemias can be ameliorated by pharmacologic doses of L-leucine (*Jaako et al., 2012*; *Payne et al., 2012*), which activates mTORC1 signaling through S6K and promotes translation. Thus, our finding that RBCs depend critically on signaling through the mTORC1 pathway is consistent with the broader view that RBCs are uniquely sensitive to perturbations that affect protein translation.

A reduced activity of mTORC1 is known to reduce the size of many cell types. Consistent with this, a reduced level of mTORC activity leads to a microcytic anemia either as a result of iron deficiency or a red cell specific mutation in Raptor. As mentioned, HRI, another iron-dependent kinase, is also known to play a critical role in mediating the response to iron deficiency (*Kazemi et al., 2007*). Further studies may reveal whether the roles of this kinase and mTORC1 in the response to iron deficiency are redundant or if there is cross-talk between these two pathways. Our data further indicate that increased mTORC1 activity in animals with a red cell specific TSC mutation can increase RBC size, similar to the effects of B12 and folate deficiency. Further studies may reveal whether a deficiency of these nutrients leads to mTORC1 activation or whether a different mechanism contributes.

A surprising finding from our study was that S6K signaling in RBCs is rapamycin resistant yet remains sensitive to ATP competitive mTOR inhibitors. Countless studies have shown that rapamycin potently blocks S6 phosphorylation in diverse cellular contexts, and the recent mTOR crystal structure has elucidated the structural origin of this sensitivity (*Yang et al., 2013*). The biochemical basis for the rapamycin resistance of RBCs is unclear, but it is intriguing that pharmacokinetic studies have reported that rapamycin preferentially accumulates in RBCs (*Yatscoff et al., 1995*; *Yatscoff, 1996*). This has been proposed to result from the elevated expression of (unidentified) rapamycin binding proteins in the erythrocytes. It is thus possible that RBCs express an unusual complement of FKBPs that blunt rapamycin's ability to block mTORC1 signaling. This would provide an explanation for why the widespread clinical use of rapamycin as an immunosuppressant is not associated with a concomitant impairment of erythropoiesis.

There is considerable interest in the potential of mTOR inhibitors for the treatment of cancer. While rapamycin analogs have been used clinically for more than a decade, the first ATP competitive mTOR inhibitors are currently in the early-stage clinical trials. A key question for these drugs is how their side-effect profile will compare to widely used rapalogs. Some studies have shown that ATP-competitive mTOR inhibitors actually have weaker effects on lymphocytes than rapamycin, suggesting that these drugs may be safer and less immunosuppressive (*Janes et al., 2010*). Our data by contrast indicate that ATP competitive mTOR inhibitors block erythropoiesis much more potently than rapamycin, particularly under conditions of erythropoietic stress. Consistent with these data, anemia has been identified as a dose-limiting toxicity in some patients enrolled in the phase I trial of MLN0128 (*Infante et al., 2012*). As impairment of erythropoiesis is a common side effect of many chemotherapies, it will be important to monitor how these drugs interact with ATP competitive mTOR inhibitors that potently block RBC function. These data also suggest the need for further studies of patients on these medications in clinical settings where hematopoiesis may be disrupted.

The results reported here demonstrate how the transcriptional profiling of activated cells can reveal the cell types that respond to specific stimuli. A basic challenge for biology is to understand how complex physiologic processes emerge from the action of individual cell types, genes, and pathways. Advances in mouse genetics have made it possible to test hypotheses about the function of genes in specific tissues using methods such as Cre-lox recombination. Yet, the existence of thousands of distinct cell types in mammals suggests a need for new approaches to prioritize and guide this effort. In this paper and a companion report (*Knight et al., 2012*), we have described a new way to map biochemical events onto cell types within a complex tissue. Our approach is based on capturing RNA from cells in proportion to a biochemical signal such as mTORC1 activation, and then using RNA sequencing to quantify this signal across numerous cell-type-specific marker genes in parallel. We have previously used this technique to identify neurons in the mouse brain that are activated during a behavior (*Knight et al., 2012*). In this study, we describe how this same approach has revealed a critical role for mTORC1 signaling in a hematopoietic cell type that was only a trace contaminant in our original experiments. These studies illustrate how unbiased genomic approaches such as RNA sequencing can be powerfully repurposed to measure the functional activity of cell types in a complex tissue. We believe the broader application of these approaches has the potential to transform how biologists monitor and analyze complex physiologic systems.

## Materials and methods

### Materials

*Vav-iCre* (#008610), *hCD2-iCre* (#008520) *Tsc1^{fl/fl}* (#005680), and *Rptor^{fl/fl}* (#013188) mice were obtained from Jackson laboratory. Rabbit anti-pS6 235/236 (Cell Signaling, Danvers MA #4858, 1:1000), rabbit

anti-pS6 240/244 (Cell Signaling #5364, 1:1000), mouse anti-total S6 (Cell Signaling #2317, 1:500), rabbit anti-p4E-BP1 Ser65 (Cell Signaling #9456, 1:500), rabbit anti-4E-BP1 rabbit anti-alpha globin (Epitomics, Burlingame, CA #EPR3608, 1:5000), rabbit anti-neuron-specific enolase (Immunostar, Hudson, WI #22521, 1:100), and HRP-conjugated rabbit anti-actin (Cell Signaling #5125, 1:2500) were used for immunoblotting. KU-0063794 was from Selleckchem (Houston, TX), MLN0128 was from Active Biochem (Maplewood, NJ), phenylhydrazine was from Sigma (St. Louis, MO), and rapamycin was from LC Labs (Woburn, MA).

## Cell culture

K562 cells were grown in RPMI supplemented with 10% dialyzed FBS. Cells were treated for 2 hr with either deferoxamine (30 µM), deferasirox (50 µM), or vehicle (0.05% DMSO). Cells were then collected by centrifugation, washed, and lysed in a 1% NP40 buffer.

## Hypothalamic RNA analysis

The hypothalamic ribosome profiling experiment that revealed enrichment of hba-a1 and hbb-b1 has previously been described (*Knight et al., 2012*). For perfusion experiments, control mice were killed by cervical dislocation and the hypothalamus was dissected. Perfused mice were anesthetized with isoflurane, transcardially perfused with PBS for 5 min, and then the hypothalamus was dissected. Hypothalamic homogenates were prepared as previously described (*Knight et al., 2012*), and total RNA was purified using the RNeasy Micro Kit. The indicated transcripts were then quantified by qPCR using internally quenched probes (Integrated DNA Technologies, Coralville, IA) with the Taqman Gene Expression Master Mix (Applied Biosystems, Foster City, CA). Values were normalized to the abundance of rpl23.

## Low iron diet

Mice were maintained on either a standard chow diet containing 220 ppm iron (Purina 5015) or a low iron diet containing 2–6 ppm iron (Harlan, TD80396). Mice were additionally given subcutaneous injections of deferoxamine (150 mg/kg) or vehicle (HBSS) 5 days a week. Reticulocyte lysates for western blotting were generated by collecting blood in EDTA capillaries by cardiac puncture and diluting into HBSS + 20 mM EDTA. This blood was pelleted (3 min at 3000 rpm at 4°C), washed three times with HBSS/EDTA, and then subjected to hypotonic lysis on ice for 20 min with occasional mixing. This was then centrifuged (5 min at 3000 rpm at 4°C), and the supernatant was collected and used for western blotting.

## Antibody staining and flow cytometry

Freshly isolated spleens were mechanically dissociated and strained through a 70-µm strainer in the presence of cold staining buffer (0.2% BSA and 5 mM glucose in PBS). Bone marrow cells were isolated from the femur using cold buffer. Fetal liver cells were extracted from E14.5 embryos, dissociated mechanically by pipetting in cold buffer. All cells were washed twice and resuspended to approximately $1.5 \times 10^6$ cells/200 µl. Cells were then stained with PE-Cy-conjugated Ter119 and PE-conjugated CD71 at a concentration of 2.5 µg/ml for 20 min at 4°C in the dark. Cells were washed and resuspended in staining buffer (1 mL) for flow cytometry. Control samples included unstained cells and single-color controls to calculate compensation. Cells were analyzed on an LSRII (BD Biosciences, San Jose, CA) flow cytometer and the data were analyzed with FlowJo software.

## Retic counting

To determine the percentage of reticulocytes in the blood, peripheral blood (5 µl) was collected in EDTA-coated capillary tubes and diluted into 1 mL of BD Retic Count/Thiazole Orange Reagent (BD #349204). Cells were stained in the dark at room temperature for 30 min. Control (unstained) samples were diluted into 1 ml buffer (0.1% BSA and 1 mM EDTA). Samples were analyzed on a LSRII (BD Biosciences) flow cytometer and the data were analyzed with FlowJo software.

## Complete blood counts

Complete blood counts were performed on peripheral blood collected using EDTA-coated capillaries. Peripheral blood was diluted 10-fold into cold buffer (HBSS containing 20 mM EDTA) and subjected to automated analysis using a Bayer Advia120 hematology analyzer.

## Drug treatment

Animals were given daily intraperitoneal injections of either rapamycin (10 mg/kg) or twice daily injections of INK-128 (2 mg/kg) or vehicle. All drugs were formulated in a solution of 5% PEG400 + 5% Tween80 in PBS and delivered in a total volume of 200 µl.

## Acknowledgements

This work was supported by the JPB Foundation (JMF), NIH grants DK083531 (ZAK), and DK041096 (JMF). ZAK is a New York Stem Cell Foundation—Robertson Investigator and acknowledges support from the McKnight Endowment Fund, the Klingenstein Fund, the Brain and Behavior Research Foundation, and the Program for Breakthrough Biological Research.

## Additional information

### Funding

| Funder | Grant reference number | Author |
|---|---|---|
| Howard Hughes Medical Institute | | Jeffrey M Friedman |
| National Institutes of Health | DK083531 | Zachary A Knight |
| National Institutes of Health | DK041096 | Jeffrey M Friedman |
| JPB Foundation | | Jeffrey M Friedman |

The funders had no role in study design, data collection and interpretation, or the decision to submit the work for publication.

### Author contributions

ZAK, SFS, Conception and design, Acquisition of data, Analysis and interpretation of data, Drafting or revising the article; KB, KT, Conception and design, Acquisition of data, Analysis and interpretation of data; JMF, Conception and design, Analysis and interpretation of data, Drafting or revising the article

### Ethics

Animal experimentation: All procedures were carried out in accordance with the National Institutes of Health Guidelines on the Care and Use of Animals and approved by the Rockefeller University Institutional Animal Care and Use Committee (Protocol #12530).

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
