## [Decision Letter]

Thank you for sending your work entitled “Ribosome profiling reveals a critical role for mTORC1 in erythropoiesis and anemia” for consideration at *eLife*. Your article has been favorably evaluated by a Senior editor and 3 reviewers, one of whom is a member of our Board of Reviewing Editors.

The following individuals responsible for the peer review of your submission have agreed to reveal their identity: David Ginsburg (Reviewing editor); Stuary Orkin and Kun-Liang Guan (peer reviewers).

The Reviewing editor and the other reviewers discussed their comments before we reached this decision, and the Reviewing editor has assembled the following comments to help you prepare a revised submission.

This manuscript demonstrates a clear role for the mTORC1 pathway in erythropoiesis. It has been evident for a long time that a high level of protein synthesis is critical to producing hemoglobin in the quantities required in the red cell, and mTOR, which positively promotes translation, has been shown to be inhibited by iron deficiency in RBCs, suggesting a mechanism of coordination between hemoglobin translation and iron availability. However, the specific link and roles of mTORC1 and mTORC2 have not been previously investigated. The authors' findings demonstrate that abnormally high or low mTORC1 activity induces macrocytic or microcytic anemia, respectively. Furthermore, pharmacological inhibition of mTOR sensitizes mice to hemolytic stress. Moreover, this work yields some surprising findings regarding attenuated effects of rapamycin on S6K signaling (versus effects of MLN0128, an ATP competitive mTOR inhibitor) that are of chemotherapeutic relevance. These findings arose from a serendipitous finding of globin RNAs in rpS6 enriched ribosomes. This indicates the potential sensitivity of this method for identifying rare populations with enhanced mTORC1 activity. The experiments are carefully performed, the data of high quality, and the results support the conclusions.

1) Does S6 phosphorylation via the mTOR pathway result in altered expression of other erythroid genes in addition to alpha and beta hemoglobin? If available, it would be interesting to see a more complete analysis of mRNAs enriched by pS6 pulldown analyzed with RNAseq. Are all rbc mRNAs similarly selected, or is there a differential effect on a specific subset of mRNAs?

2) What is the cause of death in TSC-KO mice on an iron deficient diet? The degree of anemia observed in these animals does not seem to be sufficient to explain this mortality.

3) Vav-Cre tissue specific raptor knockouts die perinatally, apparently as a result of a profound defect in erythropoiesis. Though selective knockout of raptor in the lymphoid compartment does not affect survival, is there evidence for a defect in any other hematopoietic compartment (e.g. the megakaryocyte or granulocyte/macrophage lineage)?

4) The authors chose Vav-cre mice to reveal the conditional phenotypes, because of variegated effects and incomplete recombination with other erythrocyte or hematopoietic promoters (Results section). However, an erythropoietin receptor (EpoR)-cre line (Blood 104: 659-66, 2004) provides quantitative and exquisite erythroid-specific excision and would have obviated the use of hCD2-Cre mice to exclude potential effects of lymphoid cells. Nonetheless, given the length of time and expense of repeating experiments with this strain, additional experiments of this kind should not be required, given the clear findings in the manuscript.

5) Figures 1 and 2: S6 is not a direct substrate of mTORC1. In order to strengthen the evidence for mTORC1 activation, the authors should perform Western blotting for S6K1 and 4EBP1, both are direct mTORC1 substrates.

6) Figures 3 and 4: Does the ratio of hemoglobin/total protein change? These data should be presented. Such information will show whether mTORC1 is selectively involved in hemoglobin translation.

7) Is mTORC1 activated during erythropoietic stress? This can be easily done by Western blotting of pS6 or pS6K1.

---

## [Author Response]

*1) Does S6 phosphorylation via the mTOR pathway result in altered expression of other erythroid genes in addition to alpha and beta hemoglobin? If available, it would be interesting to see a more complete analysis of mRNAs enriched by pS6 pulldown analyzed with RNAseq*. *Are all rbc mRNAs similarly selected, or is there a differential effect on a specific subset of mRNAs?*

We now include a complete list of the genes that were enriched after the polysome precipitation. In response to the reviewers’ question, we do not see enrichment for eyrothoid catalase, carbonic anahydrase II , two cytoplasmic proteins, or sprectrin-a, spectrin-b, and ankyrin, which are membrane proteins.

We assume that this is a result of the much greater abundance of hemoglobin compared to these other genes. This is noted in the revised manuscript.

*2) What is the cause of death in TSC-KO mice on an iron deficient diet? The degree of anemia observed in these animals does not seem to be sufficient to explain this mortality*.

We have performed autopsies of two of the TSC-KO mice at 6 months of age (a time after the hematologic analyses that were performed), one on a chow diet and the other on a low iron diet with an iron chelator. As now mentioned in the revised paper, pathologic analyses showed extensive extra-medullary hematopoiesis and phagocytosis of RBCs, which was greater in the iron deficient animal. At six months of age both animals also showed a widely disseminated histiocytic sarcoma in multiple sites. Consistent with this a prior report showed histiocytic sarcoma with increased Tor activity in hematopoietic cells in animals with a Pten knockout in Mx-1 cre mice. (37) This finding is discussed in the revised manuscript.

*3) Vav-Cre tissue specific raptor knockouts die perinatally*, *apparently as a result of a profound defect in erythropoiesis. Though selective knockout of raptor in the lymphoid compartment does not affect survival, is there evidence for a defect in any other hematopoietic compartment (e.g. the megakaryocyte or granulocyte/macrophage lineage)?*

The only change we saw was an increase in the white blood cell count of the RAP-KO and saw no changes in the basophil, platelets, monocytes or any other hematopoietic cell type in these animals. These data are now included in the revised manuscript.

*4) The authors chose Vav-cre mice to reveal the conditional phenotypes, because of variegated effects and incomplete recombination with other erythrocyte or hematopoietic promoters (Results section). However, an erythropoietin receptor (EpoR)-cre line (Blood 104: 659-66, 2004) provides quantitative and exquisite erythroid-specific excision and would have obviated the use of hCD2-Cre mice to exclude potential effects of lymphoid cells. Nonetheless, given the length of time and expense of repeating experiments with this strain, additional experiments of this kind should not be required, given the clear findings in the manuscript*.

We appreciate the reviewers’ consideration.

5) Figures 1 and 2: S6 is not a direct substrate of mTORC1. In order to strengthen the evidence for mTORC1 activation, the authors should perform Western blotting for S6K1 and 4EBP1, both are direct mTORC1 substrates.

We performed these studies albeit with some difficulty owing to our inability to obtain suitable antibodies. The new data in revised Figure 6 show high levels of phosphorylation of 4EBP1 and S6 in the reticulocytes of phenylhyrazine animals that was completely abolished by treatment with MLN0128. Since 4EBP1 is a direct target of mTORC1 we hope that this addresses the reviewer’s request for data on a primary substrate. Unfortunately we were unable to find a suitable antibody for S6K1, which is another direct target.

*6)*
Figures 3 and 4*: Does the ratio of hemoglobin/total protein change? These data should be presented. Such information will show whether mTORC1 is selectively involved in hemoglobin translation*.

The abundance of hemoglobin represents such a high proportion of the total protein that we were unable to obtain interpretable data addressing this point.

*7) Is mTORC1 activated during erythropoietic stress? This can be easily done by Western blotting of pS6 or pS6K1*.

The revised Figure 6 shows high levels of pS6 and p4EBP1 in phenylhyrazine treated animals.